# Investigating host-virus interaction mechanism and phylogenetic analysis of viral proteins involved in the pathogenesis

**Ahmad Abu Turab Naqvi[1], Farah Anjum[2], Alaa Shafie[2], Sufian Badar[1], Abdelbaset Mohamed Elasbali[3], Dharmendra Kumar Yadav[4]\*, Md. Imtaiyaz Hassan[5]\***

**1** Department of Computer Science, Jamia Millia Islamia, New Delhi, India, **2** Department of Clinical Laboratory Sciences, College of Applied Medical Sciences, Taif University, Taif, Saudi Arabia, **3** Clinical Laboratory Science, College of Applied Medical Sciences-Qurayyat, Jouf University, Sakakah, Saudi Arabia, **4** College of Pharmacy, Gachon University of Medicine and Science, Hambakmoeiro, Yeonsu-gu, Incheon City, South Korea, **5** Centre for Interdisciplinary Research in Basic Sciences, Jamia Millia Islamia, Jamia Nagar, New Delhi, India

\* dharmendra30oct@gmail.com (DKY); mihassan@jmi.ac.in (MIH)

## Abstract

Since the emergence of yellow fever in the Americas and the devastating 1918 influenza pandemic, biologists and clinicians have been drawn to human infecting viruses to understand their mechanisms of infection better and develop effective therapeutics against them. However, the complex molecular and cellular processes that these viruses use to infect and multiply in human cells have been a source of great concern for the scientific community since the discovery of the first human infecting virus. Viral disease outbreaks, such as the recent COVID-19 pandemic caused by a novel coronavirus, have claimed millions of lives and caused significant economic damage worldwide. In this study, we investigated the mechanisms of host-virus interaction and the molecular machinery involved in the pathogenesis of some common human viruses. We also performed a phylogenetic analysis of viral proteins involved in host-virus interaction to understand the changes in the sequence organization of these proteins during evolution for various strains of viruses to gain insights into the viral origin's evolutionary perspectives.

## Introduction

Since the first virus was discovered in 1898 [1], humans have been curious about this microorganism that causes viral diseases in plants, animals and humans. The recent outbreak of the COVID-19 pandemic has galvanized human efforts to understand the intricate mechanisms that allow viruses to infect their hosts. The first step in this course of infection that helps viruses invade the host environment is host-virus interaction. Anti-viral drugs and vaccines have been the primary combat tools to cure and safeguard humans from disease-causing viruses.

Advances in biological sciences have increased our understanding of the mechanism by which these viruses infect their hosts and cause disease. However, we are still facing severe

**Data Availability Statement:** All relevant data are within the paper.

**Funding:** This work was supported by Taif University Researchers Supporting Project Number

(TURSP-2020/131), Taif University, Taif, Saudi Arabia. The funder provided support in the form of salaries for authors FA and AS, but did not have any additional role in the study design, data collection and analysis, decision to publish, or preparation of the manuscript. The specific roles of these authors are articulated in the 'author contributions' section.

**Competing interests:** The authors have declared that no competing interests exist.

limitations in eradicating the possible threats caused by the viruses. The recent COVID-19 pandemic has raised serious questions about the status of therapeutic intervention for viral diseases. [2–7]. Viruses with a relatively minimal amount of molecular machinery compared to other evolved organisms have killed millions of people and caused millions of dollars of economic loss all over the globe. So far, more than 5 million people have died due to COVID-related complications caused by the SARS-COV-2 [8–12].

Here, we investigated the progress made in understanding human viruses and their mode of interaction. In addition, we performed a phylogenetic analysis of the viral proteins that are involved in host-virus interaction. While searching for relevant literature about the phylogenetic analysis of viral proteins that play a crucial role in host-virus interaction, we came across several studies and reviews [13–16] that elaborate on the evolutionary aspects of the viral genomes. Thus, the spectrum of studies is broad and precise for looking for relevant information about viral evolution. Therefore, we summarize the evolutionary aspects of the most frequent human viruses using phylogenetic analysis of their viral proteins, which have pathogenic significance.

## Materials and methods

### Literature survey

We performed an extensive literature survey to understand the mechanism of virion interactions with the host cells and how far we have reached our current understanding of the host-virus mechanism. For the literature survey, the PubMed (https://pubmed.ncbi.nlm.nih.gov/) repository was searched with specific search-strings such as "human viruses," "host-virus interaction," and "host-virus interaction mechanism," etc. Then, viruses were selected for the analysis based on results retrieved from the literature survey and the extent of viral disease prevalence in humans. Table 1 summarises these viruses and the different types and diseases they cause.

### Selection of viral proteins

The basis for selecting viral proteins was their role in host-pathogen interactions. Those proteins involved in host receptor interaction, cell binding, and membrane fusion were selected for sequence and phylogenetic analysis.

### Sequence and phylogenetic analysis

Here, we dwell on the phylogeny of these viral proteins to bring forward aspects of evolutionary changes that these viruses follow and change their genomic blueprints to make new and more potential viral proteins. First, FASTA sequences of major proteins involved in viral pathogenesis were downloaded from the Uniport (https://www.uniprot.org/) database. Then, the multiple sequence alignment was performed using MAFFT (https://www.ebi.ac.uk/Tools/msa/mafft/). We chose the default parameters set by the MAFFT, which uses the BLOSAM62 scoring matrix for protein sequences and with a gap penalty of 1.53. Alignments retrieved from MAFFT in FASTA format were further processed using JalView (https://www.jalview.org/). MAFFT applies the neighbor-joining method for phylogenetic tree construction for the given alignments. The data for phylogenetic trees obtained from MAFFT was used to generate circular cladograms using iTOL (https://itol.embl.de/).

## Results and discussion

For the first time, the existence of viruses was noticed by Russian biologist Dmitry Ivanovsky in 1892 as "non-bacterial pathogens," as he described them, which were later identified as

**Table 1. An overview of the human viruses, their mode of transmission, pathogenesis, genome and viral proteins.**

| S. No. | Virus | Genus/Family | Transmission | Pathology | Viral Genome | Viral Proteins |
|---|---|---|---|---|---|---|
| 1. | Influenza A Virus | Alphainfluenzavirus/ Orthomyxoviridae | Respiratory | Flu | Negative Sense Single Stranded RNA | PB1, PB1-F2, PB2, PA, PA-X, HA, NP, NA, M1, M2, NS1, NEP |
| 2. | Influenza B Virus | Betainfluenzavirus/ Orthomyxoviridae | Respiratory | Flu | Negative Sense Single Stranded RNA | PB1, PB1-F2, PB2, PA, PA-X, HA, NP, NA, M1, M2, NS1, NEP |
| 3. | Hepatitis A Virus | Hepatovirus/ Picornaviridae | Faecal-oral | Hepatitis | Positive Sense Single Stranded RNA | VP1, VP2, VP3, 2B, 2C, 3A, 3B |
| 4. | Hepatitis B Virus | Orthohepadnavirus/ Hepadnaviridae | Sexual Contact, Blood | Hepatitis | Partially Double Stranded DNA | S-HBsAg, M-HBsAg, L-HBsAg, DNA Polymerase, HBx |
| 5. | Hepatitis C Virus | Hepacivirus/ Flaviviridae | Sexual Contact, Blood | Hepatitis | Positive Sense Single Stranded RNA | C, E1, E2, NS1, NS2, NS3, NS4A, NS4B, NS5A, NS5B |
| 7. | Human Immunodeficiency Virus | Lentivirus/ Retroviridae | Sexual Contact, Blood | AIDS | Positive Sense Single Stranded RNA | MA, CA, SP1, NC, SP2, P6, RT, RNase H, IN, PR, gp120, gp41 |
| 8. | Human Papillomavirus | Alphapapillomavirus/ Papillomaviridae | Sexual Contact | Skin Warts, Genital Warts, Cancer | Small Double Stranded Circular DNA | E1, E2, E3, E4, E5, E6, E7, L1, L2 |
| 9. | SARS-CoV | Betacoronavirus/ Coronaviridae | Respiratory, Contact | SARS | Positive Sense Single Stranded RNA | S, E, M, N, NSP1, NSP2, NSP3, NSP4, NSP5, NSP6, NSP7, NSP9, NSP10, NSP11, NSP12, NSP13, |
| 10. | MERS-CoV | Betacoronavirus/ Coronaviridae | Respiratory | Respiratory Illness | Positive Sense Single Stranded RNA | S, E, M, N, AP, AP4A, AP4B, AP5, PLPro, 3CLPro |
| 11. | SARS-CoV-2 | Betacoronavirus/ Coronaviridae | Respiratory | Respiratory, COVID-19 | Positive Sense Single Stranded RNA | S, E, M, N, NSP1-16 |
| 12. | Ebola virus | Ebolavirus/ Filoviridae | Zoonosis, Contact, Blood | Hemorrhagic fever | Single Stranded RNA | L, GP1, GP2, NP, VP40, VP35, VP30, VP24 |
| 13. | Zika virus | Flavivirus/ Flaviviridae | Zoonosis, Mosquito Bite, Sexual Contact, Blood | Fever, Joint Pain, Body Rashes | Positive Sense Single Stranded RNA | C, pr, prM, M, E, NSP1, NSP2A, NS2B, NS3, NSP4A, NSP4B |
| 14. | Nipah virus | Henipavirus/ Paramyxoviridae | Zoonosis, Animal Bite, Contact | Encephalitis | Single Stranded RNA | F, G, M, P, N, L |
| 15. | Dengue virus | Flavivirus/ Flaviviridae | Zoonosis, Mosquito Bite | Haemorrhagic Fever | Positive Sense Single Stranded RNA | **C, prM), E,** NS1, NS2A, NS2B, NS3, NS4A, NS4B, NS5 |
| 16. | Chikungunya virus | Alphavirus/ Togaviridae | Zoonosis, Mosquito Bite | Haemorrhagic Fever, Joint Pain, Body Rashes | Positive Sense Single Stranded RNA | C, E1, E2, NSP1, NSP2, NSP3, NSP4 |
| 17. | Rabies virus | Lyssavirus/ Rhabdoviridae | Zoonosis, Animal Bite (Mainly Dogs) | Fatal Encephalitis | Negative Sense Single Stranded RNA | N, P, M, G, L |

Tobacco mosaic viruses. This plant virus infects the tobacco leaves, hence the name, and was given in 1898 by Martinus Beijerinck. Since this milestone, discovering viruses has been a center of attraction for biologists worldwide working on these microscopic pathogens. However, human viruses were discovered later when their discovery was fuelled by yellow fever, Influenza outbreaks, etc., [17–19]. Therefore, it may be helpful to briefly discuss the basic biology of viruses before discussing their infection mechanism [20–23].

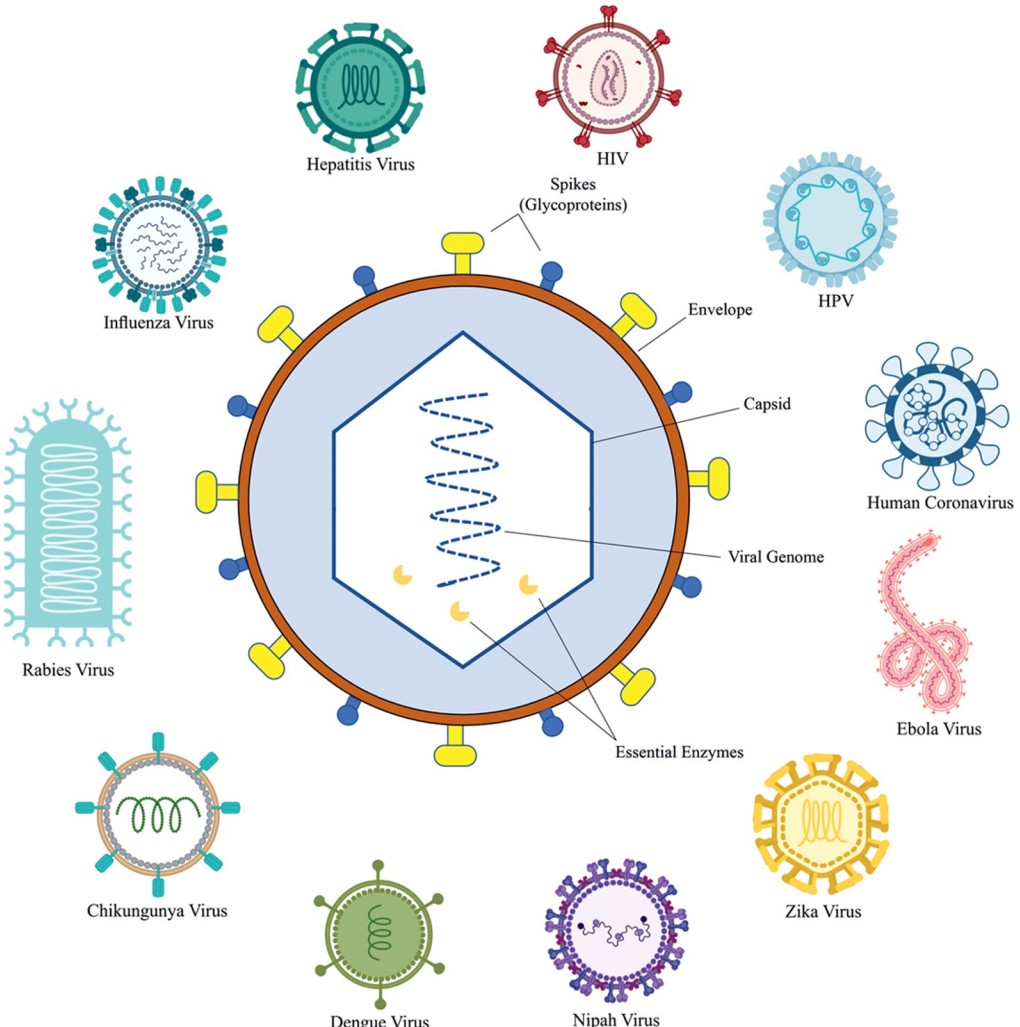

**Fig 1. Graphical representation of a typical enveloped virus (center) showing spike projections, capsid and viral genome along with the typical diagrams of human viruses (outer circle).**

Viruses are called organisms "on the edge of life" because they can only replicate and grow within a host cell [24, 25]. As a result, they rely entirely on a host body for survival. Viruses are submicroscopic particles known as virions that contain RNA or DNA as their genetic material, encapsulated in a capsid made up of capsid proteins and sometimes an outer lipid layer. The viral genome size ranges from a few kilobytes expressing only two proteins to several mega-bytes expressing up to 2500 proteins (See **Fig 1** for the typical representation of enveloped virus).

A virus reaches its host by various means [26–29]. The most notable of these is through a vector (dengue and chikungunya viruses), via infected fecal matter (gastritis), and through the blood and internal fluids of the infected person (HIV/HPV) or air via nasal/cough droplets or aerosols (Influenza or SARS viruses). It takes a virus to pass through six primary stages of infection to reach a host cell, grow and disseminate. These six stages are attachment, fusion, penetration, uncoating, replication, assembly, and release [30, 31]. **Fig 2** shows the initial steps of retrovirus host-interaction, elucidating how a virus interacts with the host and subsequently replicates and forms new virion particles released in the host body, searching for new target

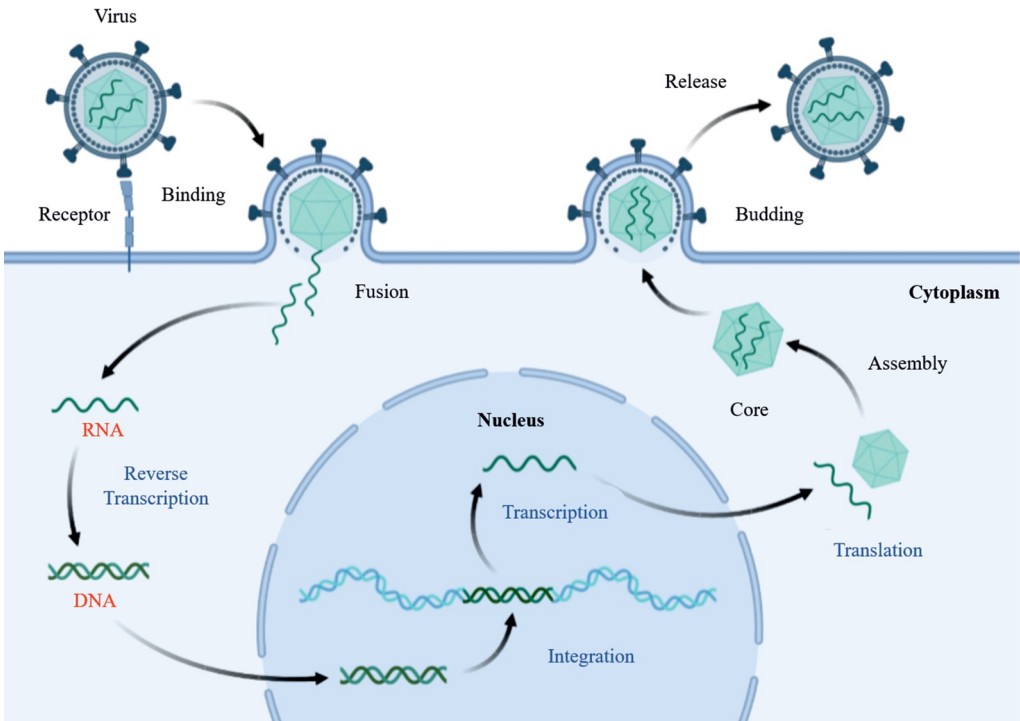

**Fig 2. Diagrammatic representation of host-virus interaction, cell entry, replication, budding and release of the newly formed virus.**

cells [32]. Therefore, it will be beneficial to briefly overview the known human viruses to develop a sense of the subject before dwelling on how viruses interact with their host and their mechanism to reach their hosts and grow themselves successfully. Table 1 gives an overview of human viruses and their disease.

## An overview of human viruses and their pathogenesis

Around 219 human viruses have been identified as causing diseases in humans ranging from mild to fatal [17, 22, 33–35]. This section provides an overview of these viruses and the diseases they cause in humans. This information is intended to lay the groundwork for subsequent sub-sections that will discuss the mode of interaction of these viruses with humans and changes in the mode of action and machinery involved in the interaction over time. Table 1 lists the most complex human viruses that cause the most lethal diseases in humans (Fig 1 shows a typical representation of human viruses discussed in this section).

 **Influenza virus.** Enveloped viruses, which are known for causing seasonal flu pandemics and epidemics, contain ssRNA as their genetic material [36]. Influenza viruses are classified into four types: influenza A, B, C, and the newly discovered type D [37]. Influenza viruses A and B are responsible for most flu cases worldwide. The difference between two surface glyco-proteins, hemagglutinin (H1 to H18) and neuraminidase, further divides influenza A virus into subtypes (N1 to N11). Out of 198 possible combinations of Influenza A subtypes, only 138 are found in nature. Following the devastating spread of the Spanish flu in 1918, the H1N1 Influenza A virus caused a pandemic in 2009, killing between 151,700 and 575,400 people worldwide. Furthermore, the annual death toll from seasonal influenza is estimated to be between 290,000 and 645,000 people worldwide [38]. The genome of the Influenza A virus contains eight negative sense segments of ssRNA.

In total, 11 proteins are encoded by eight genes, in addition to surface glycoproteins and structural proteins. Hemagglutinin and neuraminidase are two proteins required by the virus for survival and replication. The Influenza A virus, in particular, uses its hemagglutinin glycoprotein to attach to the host cell. The HA1 and HA2 subdomains aid receptor binding and cell fusion, respectively [39]. The hemagglutinin HA1 monomer's receptor binding site mediates virus anchoring by attaching to the sialic acid residues of the glycoconjugates present on the host cell surface. The process is then aided by neuraminidase, which assists hemagglutinin in scanning the host cell surface and locating the appropriately isolated receptor [40]. The successful binding and fusion of the virion particle to the host cell leads to the more important and intricate viral infection process, including the transport of the viral genome to the host cell nucleus. Discussing all the processes in detail is beyond the scope of this review. Readers may refer to these articles to better understand the underlying mechanisms [41–44].

Influenza A virus is genetically diversified due to the two main viral proteins, hemagglutinin and neuraminidase [45]. These two genes, as observed, have higher mean substitution rates, which are 5.34E−03 for hemagglutinin and 5.21E−03 for neuraminidase [46]. The receptor-binding site of hemagglutinin of the H1N1 strain of influenza A virus is found on the upper distal tip of the receptor-binding domain, which comprises amino acids 111 to 265 [47, 48]. Mutations at the receptor-binding site of the HA1 domain of hemagglutinin broadly affect its specificity and efficiency [49]. Tumpey et al., [50] have demonstrated the effects of amino acid substitutions in the HA1 domain of 1918 Influenza A virus strain, where changes resulted in the switching of binding specificity from human alpha-2,6 receptors to alpha-2,3 sialic acid receptors of the birds. It further resulted in the abolished transmission from air droplets. Vaccine development against Influenza A virus mainly targets the HA1 domain binding site to check the process of receptor binding and membrane fusion [51, 52]. As mentioned, neuraminidase also plays a significant role in viral infection, besides budding, releasing, and disseminating newly formed viral particles in the infected cells [53]. It plays a crucial role in helping the virus target cells by cleaving the sialic acids from the respiratory tract epithelial cells [54]. Neuraminidase is also known for regulating the activity of hemagglutinin. It helps to remove the oligosaccharides clouding the receptor binding site by cleaving the neuraminic acid residues of those oligosaccharides [55]. Some known neuraminidase inhibitors such as zanamivir, oseltamivir, and peramivir have been developed for Influenza infections that restrict its activity [56].

**Hepatitis virus.** A group of liver infecting viruses causing viral hepatitis and hepatocellular carcinoma consists mainly of five genetically varying (apparently unrelated) viruses: hepatitis A, B, C, D, and E. Of these, hepatitis B viruses contain partially double-stranded DNA in their genomes, while others have single-stranded RNA (ssRNA) as their genetic material [57–59]. Among the types of hepatitis virus type, A and C are of great significance due to their association with chronic liver infection and liver cancer. The Hepatitis B viral genome is enclosed inside an outer lipid envelope and an icosahedral core, which surrounds the genome and includes a DNA polymerase required for virus replication inside the host body. Encoding of four overlapping ORFs is an exciting feature of the virus. The S ORF's surface envelope proteins encoded by HBsAg are responsible for virus binding to the host cell. ORF C of the viral genome encodes HBcAg, a structural protein involved in capsid formation and replication that contributes to the viral infection. Another significant non-structural HBx protein is involved in viral infection.

The ssRNA containing the hepatitis C virus causes hepatitis C, hence its name, and is also associated with hepatocellular carcinoma. This virus is found with a relatively small molecular machinery with two glycoproteins E1 and E2, embedded in the outer lipid membrane that also help in viral binding and cell entry [60]. Surface glycoproteins E1 and E2 are formed due to

the cleavage of a precursor polyprotein [61, 62]. E1 and E2 are found in a non-covalent hetero-dimeric form on the virus surface, determining factors mediating the virus entry and pathogenicity [62]. Besides, the genetic material is enclosed inside an icosahedral capsid [63]. The viral genome encodes a single polyprotein that further breaks to produce 10 different polypeptides that play a significant role in viral replication and assembly [64].

Hepatitis B virus surface antigen HBsAg plays a significant role in initiating the viral infection and developing antibodies against HBsAg have successfully prevented the Hepatitis B virus infection (Wasley et al. Another protein other than HBsAg besides its role in viral infection and proliferation is HBx which has been extensively studied for its association with Hepatitis B virus-mediated liver cancers [65]. HBx proteins are relatively more highly expressed in hepatocellular carcinomas than other viral proteins [66]. HBx proteins promote hepatocarcinogenesis by meddling with various cellular processes and pathways such as p53 inhibition [67], dysregulating centrosome formation [68], inhibiting apoptosis by activating the p38/MAPK pathway and increasing the expression of the surviving antiapoptotic protein [69], dismantling DNA repair process [70], and by activating Jak1/STAT signaling pathway [71], etc.

**Human immunodeficiency virus.** Although two types of the human immunodeficiency virus (HIV) are known, HIV1 and HIV2, the lesser infectivity and low transmission of HIV2 are confined to only West African land [72]. Since HIV1 is more virulent and highly transmittable, it is ubiquitously known as HIV rather than HIV1. HIV can attack the host immune cells and sabotage the host immune system to evade the immune response. It mainly targets the immune system's cellular machinery, such as CD4+ T cells, dendritic cells, and other macrophages [73]. Unlike sudden pandemic outbreaks of viral infections such as the recent coronavirus pandemic, HIV has been causing a pandemic-like situation since its first emergence in 1981 and has killed an estimated 25 million people worldwide. HIV is a retrovirus with a different virion structure. The HIV genome contains two copies of ssRNAs enveloped inside a conically shaped capsid made of p24 proteins. The genome encodes nine genes that produce the molecular machinery needed to spread the virus inside the host cell.

The capsid also contains the enzymes such as reverse transcriptase, ribonuclease, integrase, and proteases, etc., needed for the replication and progression of viral particles. Envelope glycoproteins GP120 and GP41 also play a significant role in virus structure assembly. GP120 is an essential target for HIV vaccine development as it anchors the virus to human CD4 cells [74]. The most essential and foremost step towards HIV infection, which is completed in various intricate cellular processes, is the attachment of virion particles to the target cell, which is achieved by various nonspecific processes [75]. Where the viral envelope comes near the host cell via interaction either with α4β7 integrin [76] or dendritic cell-specific intercellular adhesion molecular 3-grabbing non-integrin (DC-SIGN) [77, 78]. Getting closer to the target cell surface facilitates the viral binding with the receptor CD4 protein via its interaction with the heterodimeric complex of GP41 and GP120 glycoproteins where GP120 is involved in receptor binding and also plays a significant role in host immune response evasion because of its variable loops found on the glycoprotein surface [75]. Other essential processes follow the successful receptor binding for a viral infection, such as cofactor binding, membrane fusion, viral entry, replication, release, etc.

**Human papillomavirus.** Due to its ubiquitous and frequent infection, human papillomavirus (HPV) is a grave concern for humankind. HPVs are commonly associated with genital infections that result in genital warts and, in some cases, precancerous lesions [79–81]. Unsafe sexual interactions are the most common cause of HPV transmission. HPVs are also known to contribute to and increase the risk of various cancers in humans, primarily cervical and oropharyngeal cancers [82–84].

The genome of HPV is made of a small circular dsDNA [85]. Two main culprit oncoproteins, namely E6 and E7, help the virus evade the immune system and proliferation and ultimately survive the virus and role in tumorigenesis of cervical cancers [86, 87]. These oncoproteins can inactivate tumor suppressor proteins p53 and pRb [88], where E6 is associated with the inactivation of p53, whereas E7 inactivates pRb. E6 and E7 have also been found to activate WnT signaling pathways in the cancers induced by HPV infection [89]. E6 and E7 oncoproteins dysregulate the proliferation and apoptosis in HeLa cells [90], which are well-known cell lines for studying cervical cancers. These viral oncoproteins have been broadly studied as potential hallmarks of cervical cancer therapy [91].

**Human coronavirus.** The recent outbreak of the COVID-19 pandemic from Wuhan province of China has shackled the world, causing severe damage to the world population and economy, which is still persistent and threatening the nations despite the fast development of various vaccines claiming to be up to >90% efficacy against the virus [92–94]. Human coronavirus has been linked to mild to severe respiratory tract infections in humans. Among other types of coronaviruses that infect birds and other mammals, SARS-CoV, MERS-CoV, and the novel SARS-CoV-2 have been a severe threat to human beings. Besides, HCoV-NL63 and HCoV-HKU1 have been evident in humans before the emergence of these three newfound human coronaviruses. As discussed, SARS-CoV-2 is responsible for the worst pandemics in human history, killing nearly 2.4 million people globally. The genome of the human coronavirus is made up of ssRNA that encodes various structural and non-structural genes. Envelope protein (E), membrane protein (M), and spike protein (S) embedded in lipid bilayers aid the virus in host entry and assimilation [94]. S protein is responsible for virus binding and cell entry into a host body. In addition, it interacts with human ACE2 receptors to allow the virion interaction with the host cell. Due to its significant role in host-virus interaction, spike protein has been widely studied for drug/vaccine development for COVID-19 treatment.

**Ebola virus.** The ssRNA genome of the Ebola virus encodes seven genes essential for the survival and growth of the virus [95]. Furthermore, among all the structural (seven) and non-structural proteins (three), two glycoproteins, Gp1 and Gp2 are responsible for infection by helping the virus in host cell binding and cell entry [96, 97]. Thus, the Ebola virus is well equipped with proteins capable of blocking the host cells' interferon immune system, eventually leading to a successful viral infection that is often lethal [96].

**Zika virus.** The ssRNA genome of the Zika virus is found enclosed in an icosahedral capsid. The genome encodes three structural and seven non-structural proteins in the form of an un-cleaved polyprotein, which further breaks down its components [98]. Flavivirus glycoprotein encapsulates the virus and binds the virion to the endosomal host cell membrane [99].

**Nipah virus.** The ssRNA genome of the Nipah virus is found enveloped inside a lipid bilayer. It produces six proteins that help the virus infect the host and replicate. Two glycoproteins, namely fusion glycoprotein (F) and attachment glycoprotein (G), are responsible for host cell binding and entry inside the host cell [100]. In addition, the glycoproteins interact with ephrin B2 and B3 of a human host cell for viral binding [101]. Besides, matrix protein (M), phosphoprotein (P), nucleocapsid (N), and a long polymerase (L) are also essential molecular machinery that helps the virus to increase [102].

**Dengue virus.** The ssRNA Dengue virus genome encodes a large precursor polyprotein which is comprised of three structural proteins, namely capsid protein (C), membrane protein (prM), and an envelope protein (E). It also contains seven non-structural proteins: NS1, NS2A, NS2B, NS3, NS4A, NS4B, and NS5 [103, 104]. E protein of the dengue virus is essential for its attachment to the host cell [105] by interacting with heparan sulfate receptors highly susceptible to viral infections [106].

**Chikungunya virus.** The chikungunya virus contains an encapsulated RNA genome with two open reading frames that encode two precursor polyproteins, one structural polyprotein (which cleaves into one capsid (C) protein, and two glycoproteins E1 and E2), and one non-structural polyprotein [107]. Glycoprotein E2 plays a significant role in viral attachment and entry to the host cell by interacting with the host cell receptors [108].

**Rabies virus.** The RNA genome found enveloped inside a nucleocapsid made of nucleo-proteins of the rabies virus encodes five essential structural proteins which help the virus in host attachment and proliferation, mainly targeting the muscular and nerve cells. Other than nucleoprotein (N), phosphoprotein (P), matrix protein (M), L protein, and glycoprotein (G) are essential structural proteins [109, 110]. The spike formed by the trimeric G protein helps the virus anchor to the host cell receptors [111].

## Phylogenetic analysis of the viral proteins involved in host-virus interaction

In the previous sections, we have discussed the structural and genomic features of various human viruses while focusing on the viral proteins that are involved in host attachment, bind-ing, membrane fusion, and cell entry help the virus infect a host cell and replicate to infect more and more adjacent cells in the host environment. As we know, these proteins are among the most promising targets for therapeutic interventions. In addition, these proteins have also been studied thoroughly for their variability in different viral strains and isolates.

As discussed, the influenza A virus has two main viral proteins that help infect the host cells and evade the immune system counteracting it. These proteins, hemagglutinin and neuramini-dase, have undergone various changes that have given rise to different genotypes of influenza viruses. Fig 3 shows multiple sequence alignments (MSAs) and phylogenetic trees of hemag-glutinin and neuraminidase taken from seven different Influenza A H1N1 subtype strains. These strains were selected randomly based on periodic and geographical distribution, includ-ing the viral host variability since we also selected two strains from Duck and Swine. The MSA of the hemagglutinin HA1 subdomain (Fig 3A) shows an overall conserved sequence; however, there are visible non-conserved amino acid positions distributed in the receptor-binding area of the HA1 subdomain. The phylogenetic tree for the HA1 subdomain (Fig 3C) shows three strains, i.e., USSR/90/1977, Brazil/11/1978, and India/6263/1980, which have very close evolu-tionary relationships with very minimal branch lengths. The other four strains on the phyloge-netic tree show that the Brevig Mission/1/1918 strain shares the same tree branches with the Duck/Alberta/35/1976 strain with a branch length difference of ~0.059. The Korea/01/2009 strain of H1N1 virus shares nodes with the New Jersey/11/1976 strain isolated from swine.

The MSA of functional residues contains the region of neuraminidase (Fig 3B). It shows rel-atively highly conserved positions where USSR/90/1977, Brazil/11/1978, and India/6263/1980 strains show 97% sequence identity, indicating their origin from the same ancestor during the evolutionary process. However, similar trends are observed on the phylogenetic tree for neur-aminidase (Fig 3D), with relatively higher branch lengths depicting a vast evolutionary time scale. Protein X (HBx) of the Hepatitis B virus is known for its role in virus-induced hepatocar-cinogenesis and has been widely studied as a potential biomarker.

The MSA of HBx protein (Fig 4A) from various genotypes (Table 2) of Hepatitis B virus shows highly conserved regions except for a few amino acid positions that are non-conserved among all the sequences. In addition, sequences from A1, A2, A3, C, D, and E genotypes share common amino acid substitutions that constitute non-conserved regions. A similar pattern is also observed on the phylogenetic tree (Fig 4B) for these genotypes of the Hepatitis B virus, indicating a common ancestor for these genotypes of the virus. Genotypes B1 and B2 share the

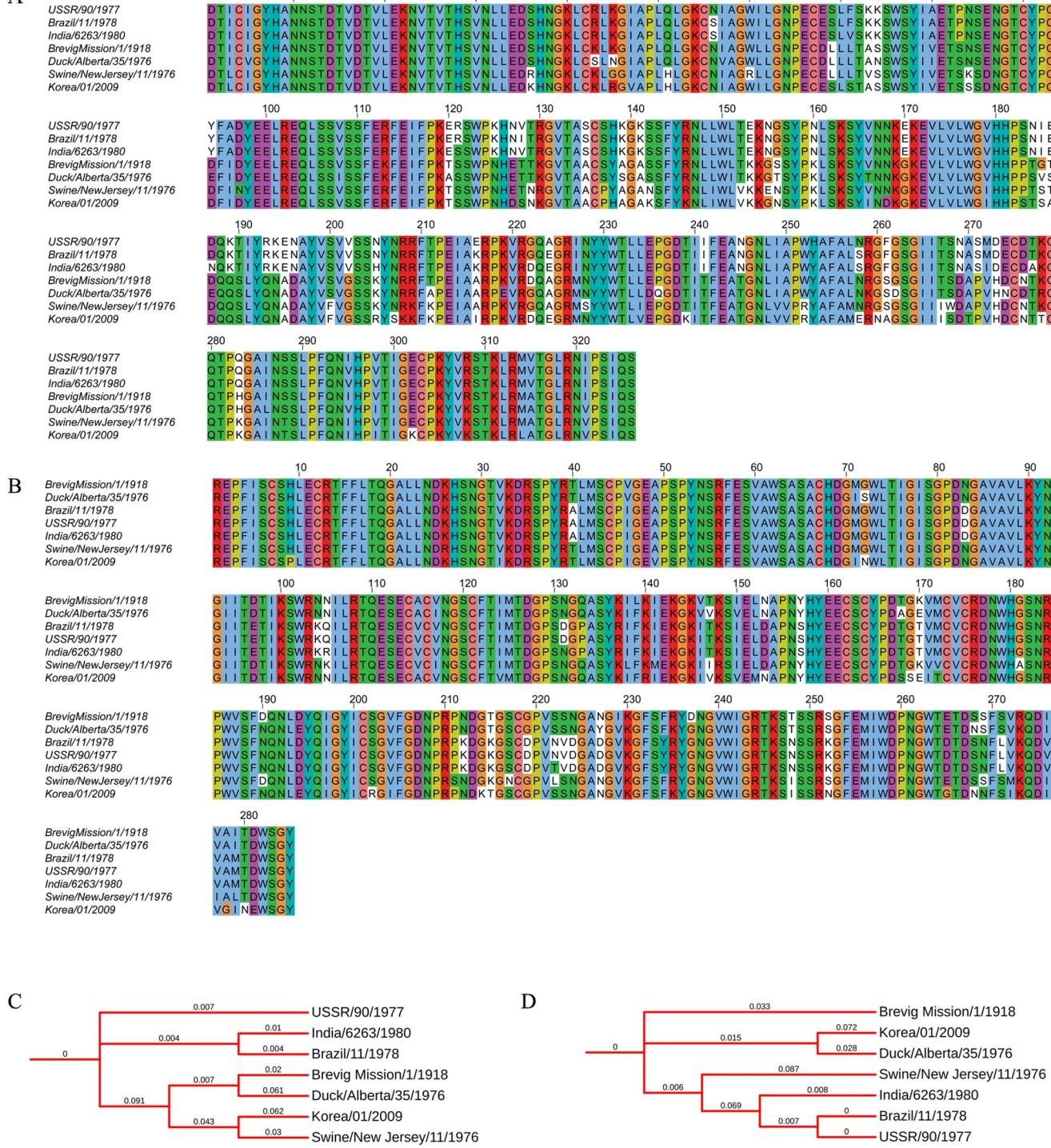

**Fig 3. Phylogenetic profile of viral proteins from H1N1 Influenza A virus.** Multiple Sequence Alignment of HA1 subdomain (A) of the hemagglutinin and active/binding site region of neuraminidase. (B) from different strains of the H1N1 Influenza A virus. (C) Phylogenetic tree of HA1 subdomain. (D) Phylogenetic tree of neuraminidase.

bifurcated branch node with these six genotypes, with branch length differences of around 0.019. Genotypes F1, F2, and H, share common nodes, while G is the most distant on the evolutionary tree with a branch length of 0.148. The E1 and E2 glycoproteins of the Hepatitis C

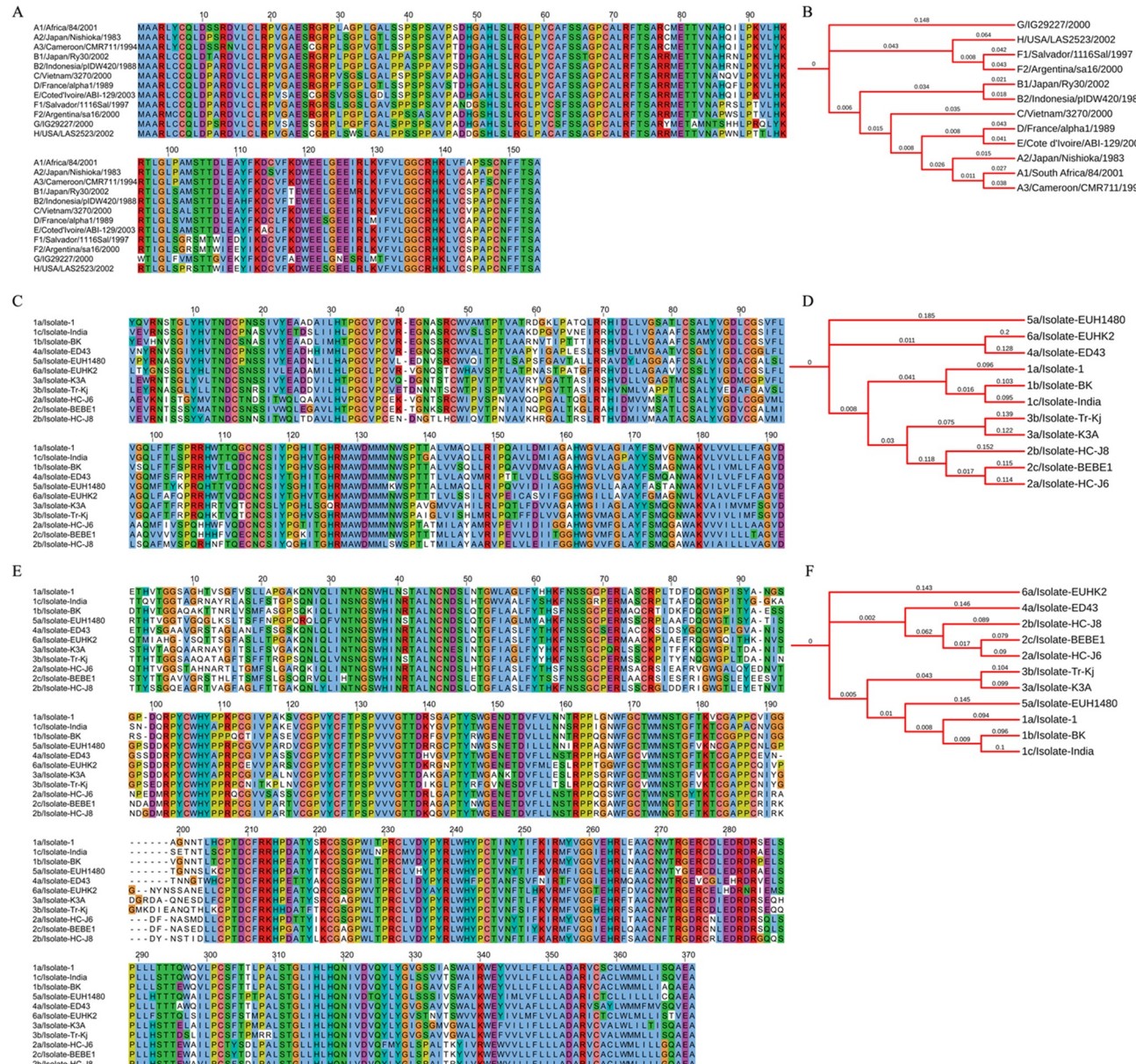

**Fig 4. Phylogenetic profile of viral proteins from Hepatitis B and Hepatitis C virus.** (A) Multiple sequence alignment of Protein X from Hepatitis B. (B) Phylogenetic tree of Protein X. (C) Multiple sequence alignment of E1 glycoprotein along with phylogenetic tree (D). (E) Multiple sequence alignment of E2 glycoprotein with the phylogenetic tree (F).

virus are significant for its survival and growth and are potential targets for vaccine development. However, the MSA of both these glycoproteins (Fig 4C and 4E) from various genotypes of Hepatitis C virus (Table 2) shows non-conserved positions spread across the sequences, including few stretches are conserved between all the genotypes. Similar patterns are observed on the phylogenetic trees (Fig 4D and 4F). The branch lengths between the parent and child nodes are relatively high, depicting a common ancestry but a higher divergence and substitution rate among the sequences.

Fig 5 shows the MSA and phylogenetic tree of GP160 glycoprotein (Surface protein GP160 and transmembrane subunit GP41) from various isolates of HIV subtypes. It is established

**Table 2. Viral Protein selected for phylogenetic analysis along with the details of virus subtype/genotype and strain/isolate.**

| S. No. | Protein | Domain/Region | Virus | Subtype/ Genotype | Strain/Isolate | UNIPROT ID |
|---|---|---|---|---|---|---|
| 1. | Hemagglutinin | HA1 Subdomain | Influenza A | H1N1 | Brevig Mission/1/1918 | Q9WFX3 (Hemagglutinin) Q9IGQ6 (Neuraminidase) |
| 2. | Neuraminidase | Active/Binding Site Residues Containing | | | Swine/New Jersey/11/1976 | P03455 (Hemagglutinin) Q9IGQ0 (Neuraminidase) |
| | | | | | Duck/Alberta/35/1976 | P26562 (Hemagglutinin) Q9IGQ1 (Neuraminidase) |
| | | | | | USSR/90/1977 | P03453 (Hemagglutinin) P03469 (Neuraminidase) |
| | | | | | Brazil/11/1978 | A4GBX7 (Hemagglutinin) A4GBY0 (Neuraminidase) |
| | | | | | India/6263/1980 | A4GCJ7 (Hemagglutinin) A4GCK0 (Neuraminidase) |
| | | | | | Korea/01/2009 | C5MQE6 (Hemagglutinin) C5MQP8 (Neuraminidase) |
| 3. | Protein X | Complete | Hepatitis B | A1 | South Africa/84/2001 | Q91C38 |
| | | | | A2 | Japan/Nishioka/1983 | P69714 |
| | | | | A3 | Cameroon/CMR711/1994 | Q4R1S1 |
| | | | | B1 | Japan/Ry30/2002 | P0C678 |
| | | | | B2 | Indonesia/pIDW420/1988 | P20975 |
| | | | | C | Vietnam/3270/2000 | Q9E6S8 |
| | | | | D | France/alpha1/1989 | P24026 |
| | | | | E | Cote d'Ivoire/ABI-129/2003 | Q80IU8 |
| | | | | F1 | El Salvador/1116Sal/1997 | Q8JMY3 |
| | | | | F2 | Argentina/sa16/2000 | Q99HR6 |
| | | | | G | IG29227/2000 | Q9IBI5 |
| | | | | H | United States/LAS2523/2002 | Q8JMY5 |
| 4. | E1 Glycoprotein | | Hepatitis C | 1a | Isolate-1 | P26664 |
| 5. | E2 Glycoprotein | | | 1b | BK | P26663 |
| | | | | 1c | India | Q913D4 |
| | | | | 2a | HC-J6 | P26660 |
| | | | | 2b | HC-J8 | P26661 |
| | | | | 2c | BEBE1 | Q68749 |
| | | | | 3a | K3A | Q81495 |
| | | | | 3b | Tr-Kj | Q81487 |
| | | | | 4a | ED43 | O39929 |
| | | | | 5a | EUH1480 | O39928 |
| | | | | 6a | EUHK2 | O39927 |

(*Continued*)

**Table 2.** (Continued)

| S. No. | Protein | Domain/Region | Virus | Subtype/ Genotype | Strain/Isolate | UNIPROT ID |
|---|---|---|---|---|---|---|
| 6. | GP160 | GP160 Surface Protein<br><br>GP41 Transmembrane Region | Human Immunodeficiency Virus | A | MAL | P04583 |
| | | | | B | ARV2/SF2 | P03378 |
| | | | | C | 92BR025 | O12164 |
| | | | | D | ELI | P04581 |
| | | | | F1 | 93BR020 | O89292 |
| | | | | F2 | MP255 | Q9QBZ4 |
| | | | | G | 92NG083 | O41803 |
| | | | | H | 90CF056 | O70902 |
| | | | | J | SE9173 | Q9WC69 |
| | | | | K | 96CM-MP535 | Q9QBY2 |
| 7. | E6 Oncoprotein | Complete | Human Papillomavirus | 16 | - | P03126(E6) P03129(E7) |
| 8. | E7 Oncoprotein | Complete | | 31 | | P17386(E6) P17387(E7) |
| | | | | 33 | | P06427(E6) P06429(E7) |
| | | | | 35 | | P27228(E6) P27230(E7) |
| | | | | 52 | | P36814(E6) P36831(E7) |
| | | | | 58 | | P26555(E6) P26557(E7) |
| | | | | 67 | | F8S5Y6(E6) F8S5U7(E7) |
| 9. | Spike Glycoprotein | Receptor Binding Domain | Coronavirus | Bat coronavirus HKU3 | | Q3LZX1 |
| | | | | Bovine coronavirus | 98TXSF-110-ENT | Q91A26 |
| | | | | Human Coronavirus HKU1 | N1 | Q5MQD0 |
| | | | | Human Coronavirus HKU1 | N2 | Q14EB0 |
| | | | | Human Coronavirus HKU1 | N5 | Q0ZME7 |
| | | | | Human Coronavirus HCoV-OC43 | | P36334 |
| | | | | MERS-CoV | United Kingdom/ H123990006/2012 | K9N5Q8 |
| | | | | SARS-CoV | | P59594 |
| | | | | SARS-CoV-2 | | P0DTC2 |

that the structural and functional variability in viral proteins, especially those involved in host receptor binding, is a significant hurdle in vaccine development. Despite decades' efforts, there is no effective vaccine against HIV yet. The contributing factor to this is the variation in the structural and functional features of the HIV viral proteins. The MSA of GP160 surface protein (Fig 5A) taken from various isolates of HIV (Table 2) shows highly variable regions in the sequences. Besides, sequence lengths also significantly vary among the subjects. A similar pattern is observed for the GP41 transmembrane subunit (Fig 5C); however, the conserved regions are comparatively higher in the GP41 subunit. Higher differences in the branch lengths of the tip nodes from the root and internal nodes on the phylogenetic trees (Fig 5B and 5D) indicate a high rate of amino acid substitutions among various subtypes of HIV. Phylogenetic analysis of E6 and E7 oncoproteins, responsible for virus-associated cervical cancers, from the human papillomavirus, shows a high divergence among HPV types 16, 31, 33, 35, 52, 58, and 67. In addition, amino acid substitution resulting in non-conserved regions is high in oncoproteins (Fig 6A and 6B). The phylogenetic tree (Fig 6C and 6D) also shows higher

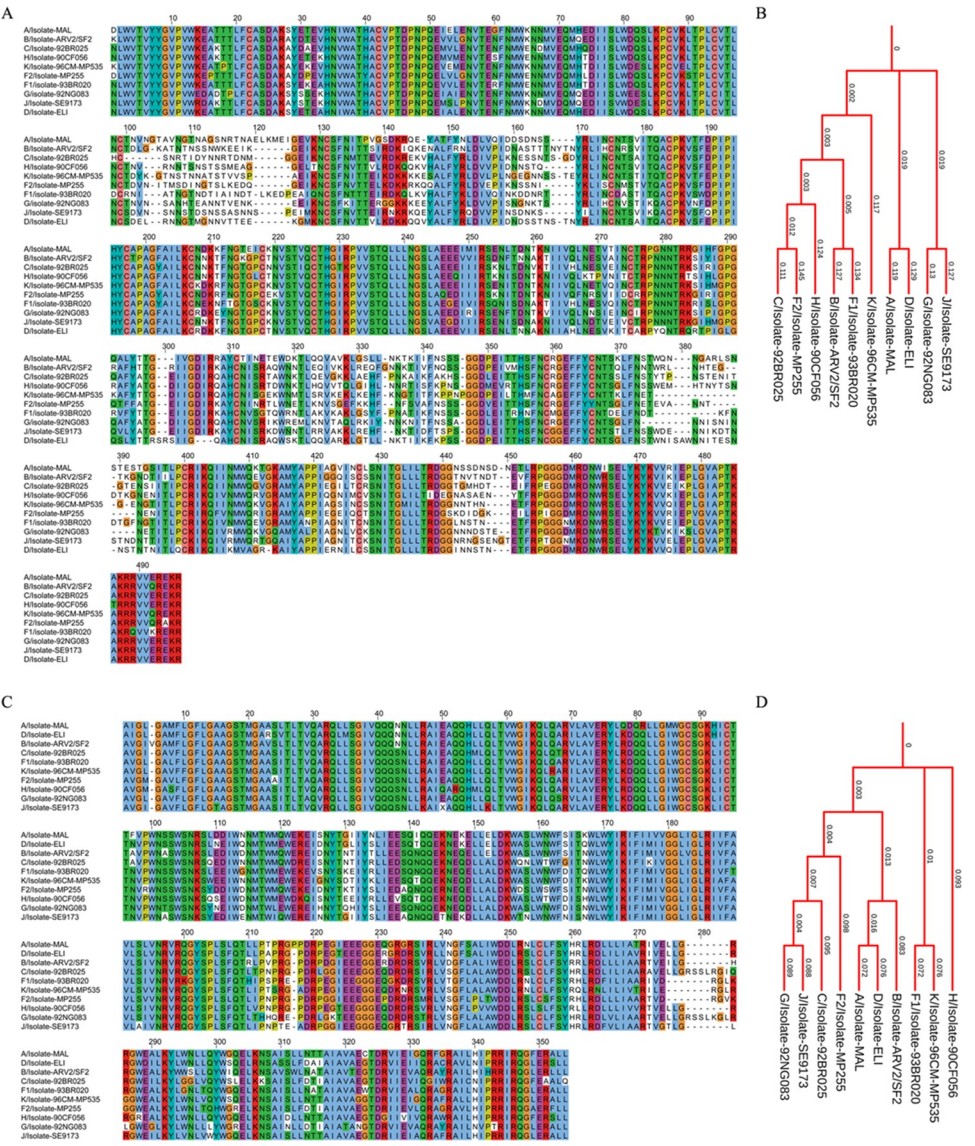

**Fig 5. Phylogenetic profile of GP160 glycoprotein from HIV.** (A) Multiple sequence alignment and (B) phylogenetic tree of surface protein region GP160. (C) Multiple sequence alignment and (D) phylogenetic tree of transmembrane region GP41.

divergence among various HPV subtypes. The spike glycoprotein of the coronavirus is responsible for interaction with the host receptor through its receptor-binding domain. The phylogenetic analysis of various coronavirus strains isolated from humans (Table 2) and other organisms (bat coronavirus and bovine coronavirus) shows regions with higher sequence variation in the receptor-binding domain of the spike glycoprotein of the coronavirus.

It is also noticeable that the sequence lengths of the receptor-binding domains vary significantly between various strains. Fig 7A shows MSA of the receptor-binding domain of Spike proteins depicting highly non-conserved regions. A spike protein is a potential target for vaccine development, and higher sequence variability might be a hurdle in the effectiveness of the vaccines under development or developed so far (Table 1). The divergence between various strains for Spike protein is very high, as observed on the phylogenetic tree (Fig 7B).

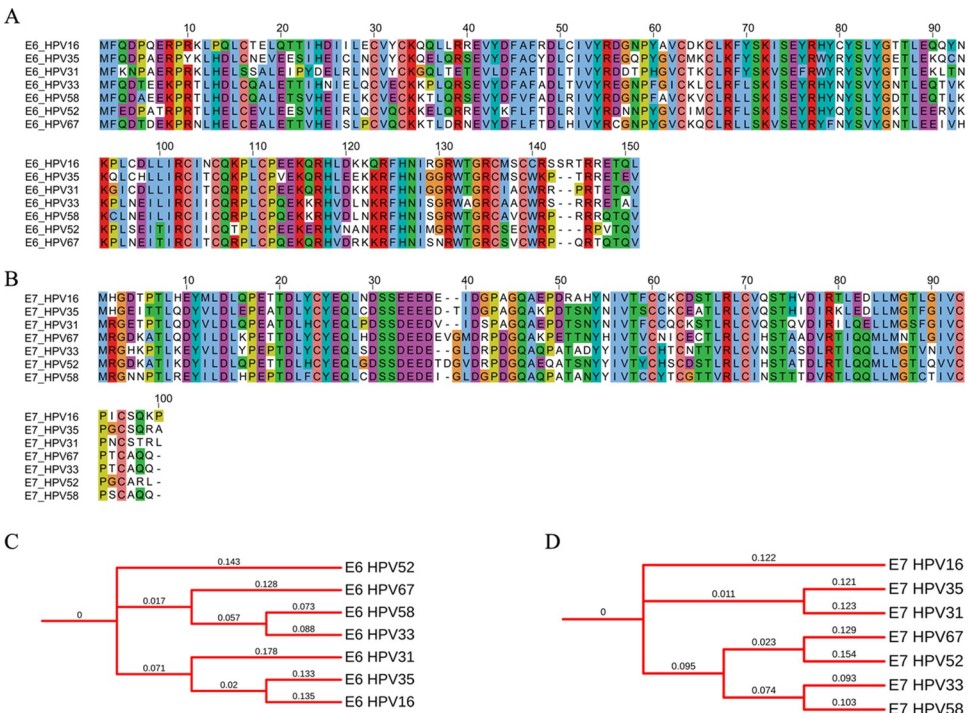

**Fig 6. Phylogenetic profile of E6 and E7 oncoproteins from HPV.** (A) Multiple sequence alignment of E6. (B) Multiple sequence alignment of E7. (C) Phylogenetic tree of E6 and (D) E7 oncoproteins.

Despite this, the propensity of non-conserved regions between SARS-CoV and the novel coronavirus SARS-CoV-2 is very high, even though they share an evolutionary origin (Fig 7B). A higher rate of amino acid substitutions indicates frequently originating viral strains that are more likely to evade immune response with a higher tendency to infect host cells. It happened in the case of the novel coronavirus UK strain, which is more lethal and highly contaminable

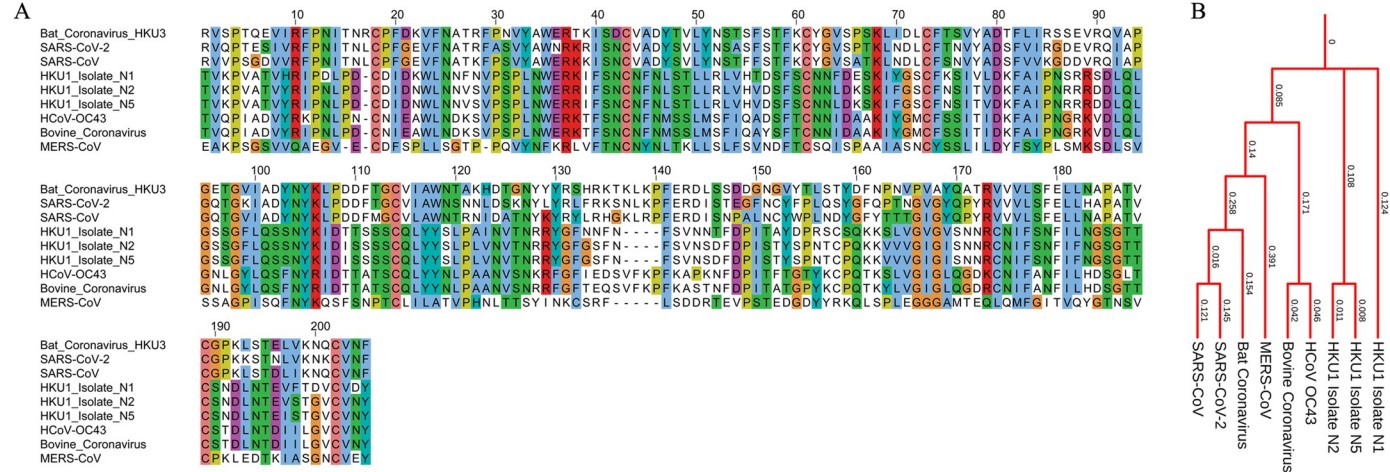

**Fig 7. Phylogenetic profile of receptor binding domain of the spike glycoprotein of the coronavirus.** (A) Multiple sequence alignment and (B) phylogenetic tree of the receptor-binding domain.

than previously known strains. Viruses tend to evolve and change their genetic makeup to evade host immune responses.

We have witnessed a remarkable change in the novel coronavirus and the origin of new viral strains that are more powerful than their progenitors. The purpose of phylogenetic analysis of significant viral proteins across strains and isolates was to understand how many changes these viruses have gone through so far. As discussed above, most of these viruses show significant changes in their host-interacting viral proteins since their discovery to current times. We also observed evolutionary changes in viral proteins taken from strains and isolates from various geographical regions.

## Conclusion

We investigated the host-virus interaction mechanisms of common human viruses and ran phylogenetic analyses of the viral proteins involved in host cell surface binding. As we have seen, viruses are well-equipped with essential molecular machinery that allows them to infect humans, replicate, and survive in the host environment. Extensive studies on the host-virus interaction mechanism have resulted in significant advances in discovering effective therapeutics for various viral diseases over the last few decades (Influenza vaccines, hepatitis vaccines, rabies vaccines, and recent vaccine development against novel coronavirus). However, in the case of some viruses, medical science is still facing challenges in the development of vaccines. The reason for this appears to be the genomic changes that these viruses go through to infiltrate the host immune response and ensure their survival. Proteins involved in host-virus interaction have significant variations in their sequence structure, as revealed by phylogenetic analysis. The majority of these variations are found in regions that contain host receptor binding motifs. The frequency and prevalence of genomic variations are a barrier to vaccine development. As a result, a thorough understanding of virus interaction mechanisms and genomic variations may aid in developing vaccines with higher efficacy against a wide range of viral strains.

## Acknowledgments

The authors thank the Department of Science and Technology, Government of India, for the FIST support (FIST program No. SR/FST/LSII/2020/782).

## Author Contributions

**Conceptualization:** Ahmad Abu Turab Naqvi, Farah Anjum, Dharmendra Kumar Yadav, Md. Imtaiyaz Hassan.

**Data curation:** Ahmad Abu Turab Naqvi, Dharmendra Kumar Yadav.

**Formal analysis:** Farah Anjum, Alaa Shafie, Sufian Badar, Md. Imtaiyaz Hassan.

**Funding acquisition:** Alaa Shafie, Abdelbaset Mohamed Elasbali.

**Investigation:** Ahmad Abu Turab Naqvi, Farah Anjum, Sufian Badar, Abdelbaset Mohamed Elasbali.

**Methodology:** Ahmad Abu Turab Naqvi, Abdelbaset Mohamed Elasbali.

**Project administration:** Abdelbaset Mohamed Elasbali.

**Resources:** Ahmad Abu Turab Naqvi, Abdelbaset Mohamed Elasbali, Dharmendra Kumar Yadav.

**Software:** Farah Anjum, Alaa Shafie, Sufian Badar, Dharmendra Kumar Yadav, Md. Imtaiyaz Hassan.

**Supervision:** Alaa Shafie, Dharmendra Kumar Yadav, Md. Imtaiyaz Hassan.

**Validation:** Ahmad Abu Turab Naqvi, Alaa Shafie, Sufian Badar, Md. Imtaiyaz Hassan.

**Visualization:** Alaa Shafie, Sufian Badar, Md. Imtaiyaz Hassan.

**Writing – original draft:** Ahmad Abu Turab Naqvi.

**Writing – review & editing:** Farah Anjum, Dharmendra Kumar Yadav, Md. Imtaiyaz Hassan.

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
