## [Decision Letter · Decision Letter 0]

8 Nov 2021

PONE-D-21-33085Investigating Host-Virus Interaction Mechanism and Phylogenetic Analysis of Viral Proteins Involved in the PathogenesisPLOS ONE

Dear Dr. Hassan,

Thank you for submitting your manuscript to PLOS ONE. After careful consideration, we feel that it has merit but does not fully meet PLOS ONE’s publication criteria as it currently stands. Therefore, we invite you to submit a revised version of the manuscript that addresses the points raised during the review process.

The manuscript has been reviewed by two independent reviewers. They find merit in this manuscript but have highlighted several areas in which improvement and corrections are necessary. These areas include the organization of the text as well as technical details and must be addressed for the manuscript to be considered for publication. Few language issues also need to be resolved by the authors.

We look forward to receiving your revised manuscript.

Kind regards,

Timir Tripathi, Ph.D.

Academic Editor

PLOS ONE

Journal Requirements:

"This work was supported by Taif University Researchers Supporting Project Number (TURSP-2020/131), Taif University, Taif, Saudi Arabia. 

We note that one or more of the authors is affiliated with the funding organization, indicating the funder may have had some role in the design, data collection, analysis or preparation of your manuscript for publication; in other words, the funder played an indirect role through the participation of the co-authors. If the funding organization did not play a role in the study design, data collection and analysis, decision to publish, or preparation of the manuscript and only provided financial support in the form of authors' salaries and/or research materials, please do the following:

a. Review your statements relating to the author contributions, and ensure you have specifically and accurately indicated the role(s) that these authors had in your study. These amendments should be made in the online form.

b. Confirm in your cover letter that you agree with the following statement, and we will change the online submission form on your behalf: 

“The funder provided support in the form of salaries for authors FA and AS, but did not have any additional role in the study design, data collection and analysis, decision to publish, or preparation of the manuscript. The specific roles of these authors are articulated in the ‘author contributions’ section.

Reviewers' comments:

Reviewer's Responses to Questions

**Comments to the Author**

1. Is the manuscript technically sound, and do the data support the conclusions?

Reviewer #1: Yes

Reviewer #2: Yes

2. Has the statistical analysis been performed appropriately and rigorously? 

Reviewer #1: Yes

Reviewer #2: Yes

3. Have the authors made all data underlying the findings in their manuscript fully available?

Reviewer #1: Yes

Reviewer #2: Yes

4. Is the manuscript presented in an intelligible fashion and written in standard English?

Reviewer #1: Yes

Reviewer #2: Yes

5. Review Comments to the Author

Reviewer #1: This paper attempts to bring and correlation between viral pathogenies with that of viral genome organization. Specifically deciphering the role of viral attachment proteins. This topic is very broad. The first half of the paper contains a very general description of the virus biology of several human pathogens. In the latter half, the authors produce phylogenetic evidence of major viral attachment protein and look for some common ground to relate it the pathogenesis.

1. In my view the paper could have a more organized approach. This manuscript contains lots of facts, general things.

2. The authors need to make it more appealing by detailing the phylogenetic information with that of virus biology cum viral pathogenesis. In a simple way, please answer how the phylogenetic information is important to predict viral pathogenesis.

3. How does the structural/viral attachment protein genetic make-up predictably results in viral entry, attachment, receptor binding, and providing an opportunity for drug discovery?

4. Working with too many viruses at a time is clearly going out of the focus. I recommend that the authors should pick a family of viruses such as flaviviruses, rhabdovirus, coronaviridae, etc goes to the species level, look at the minute details in viral attachment proteins and establish the link between the host receptor and pathogenies.

More importantly, the introduction and abstract section should highlight specific new findings, insights, or perspectives.

5. Grammar and English sentences need thorough checking,

6. Please focus on a specific group of viruses, and look at the species variation, link it to virus biology. Coronavirus shoudl be highlighted.

7. Quality of figures may be improved.

8. Please avoid passive grammars in sentence structure. References should be thoroughly checked.

Reviewer #2: In this review article, authors discuss the host-virus interaction mechanisms and the molecular machinery involved in the pathogenesis of the influenza virus, hepatitis virus, human immunodeficiency virus, and coronaviruses. They discuss the genomic organization of these viruses, including the viral proteins that play crucial roles in viral attachment, membrane fusion, cell entry, replication and subsequent release of the mature virion particles in the host environment where they infect. To gain insights into the viral origin's evolutionary perspectives, they have claimed to envisage the phylogeny of viral proteins to understand the changes in the sequence organization of these proteins during evolution for various strains of the viruses.

1. The phylogenetic analysis, which is the main highlight of the review according to the authors, seems weak.

2. Why authors have relied on summarizing viral pathogenesis in almost half of the manuscript.

3. Out of phylogenetic analysis presented, it is difficult to fetch what conclusion authors want to draw.

4. Till, they will discuss the take home messages of each of the trees and MSA representations, it is difficult to understand what evolutionary significance they provide.

6. The manuscript needs to express in the introduction/ background section comments on the current body of literature, after that, what are the gaps in our understanding of viral molecular evolution.

7. Language editing is required to improve the quality of manuscript.

6. PLOS authors have the option to publish the peer review history of their article (what does this mean?). If published, this will include your full peer review and any attached files.

Reviewer #1: No

Reviewer #2: **Yes: **RAJESH SINHA

---

## [Author Response · Author response to Decision Letter 0]

10 Nov 2021

Reviewers’ Comments

Reviewer #1: This paper attempts to bring and correlation between viral pathogenies with that of viral genome organization. Specifically deciphering the role of viral attachment proteins. This topic is very broad. The first half of the paper contains a very general description of the virus biology of several human pathogens. In the latter half, the authors produce phylogenetic evidence of major viral attachment protein and look for some common ground to relate it the pathogenesis.

1. In my view the paper could have a more organized approach. This manuscript contains lots of facts, general things.

Response 1: As per the suggestion of the review the authors have revised and updated the structure and language of the manuscript to make it more readable. 

2. The authors need to make it more appealing by detailing the phylogenetic information with that of virus biology cum viral pathogenesis. In a simple way, please answer how the phylogenetic information is important to predict viral pathogenesis.

Response 2: As suggested, changes have been made in the manuscript to elaborate the role of phylogenetic analysis and pathogenesis. 

3. How does the structural/viral attachment protein genetic make-up predictably results in viral entry, attachment, receptor binding, and providing an opportunity for drug discovery?

Response 3: This is evident from the literature available and ongoing vaccine development projects worldwide, understanding of host-virus interaction mechanism play a crucial role for the development of effective vaccines and therapeutic interventions. Therefore, we aim to shed light on these processes that help the virus to enter host cell and replicate. 

4. Working with too many viruses at a time is clearly going out of the focus. I recommend that the authors should pick a family of viruses such as flaviviruses, rhabdovirus, coronaviridae, etc., goes to the species level, look at the minute details in viral attachment proteins and establish the link between the host receptor and pathogenies.

More importantly, the introduction and abstract section should highlight specific new findings, insights, or perspectives.

Response 4: The authors have selected the various based on the prevalence of disease they cause across the globe. Mainly those viruses that have been and are associated with epidemic and pandemic like situations are discussed. As suggested, the manuscript has been revised to eliminate possible grammatical errors. 

5. Grammar and English sentences need thorough checking,

Response 5: As suggested by the reviewer, the manuscript has been revised and checked for grammatical and spelling mistakes. 

6. Please focus on a specific group of viruses, and look at the species variation, link it to virus biology. Coronavirus should be highlighted.

Response 6: The authors have selected the various based on the prevalence of disease they cause across the globe. Mainly those viruses that have been and are associated with epidemic and pandemic like situations are discussed.

7. Quality of figures may be improved.

Response 7: Figures have been improved to match the required quality measures. 

8. Please avoid passive grammars in sentence structure. References should be thoroughly checked.

Response 8: As per the suggestion of the reviewer, the manuscript has been revised to remove grammatical errors and passive sentence structure as much as possible. 

Reviewer #2

In this review article, authors discuss the host-virus interaction mechanisms and the molecular machinery involved in the pathogenesis of the influenza virus, hepatitis virus, human immunodeficiency virus, and coronaviruses. They discuss the genomic organization of these viruses, including the viral proteins that play crucial roles in viral attachment, membrane fusion, cell entry, replication and subsequent release of the mature virion particles in the host environment where they infect. To gain insights into the viral origin's evolutionary perspectives, they have claimed to envisage the phylogeny of viral proteins to understand the changes in the sequence organization of these proteins during evolution for various strains of the viruses. 

1. The phylogenetic analysis, which is main highlight of the review according to the authors, seems weak.

Response 1: As per the suggestion of the reviewer, the section discussing phylogenetic analysis has been updated to make it more elaborate. 

2. Why authors have rely on summarizing the viral pathogenesis almost half of the manuscript.

Response 2: The rationale behind discussing the pathogenesis and viral proteins involved in pathogenesis was to lay the foundation for highlighting the role of these proteins and to link their significant in the phylogenetic analysis. However, the pathogenesis section has been trimmed to reduce the redundancy. 

3. Out of phylogenetic analysis presented, it is difficult to fetch, what conclusion authors wants to draw.

Response 3: Phylogenetic analysis has been updated to make it more comprehensible. Our aim was to elaborate the evolutionary changes in the viral proteins over the span of time that play significant role in hurdles in vaccine development since most vaccine and antiviral drugs target proteins involved in cell attachment and fusion. 

4. Till, they will discuss the take home messages of each of the trees and MSA representations, it is difficult to understand what evolutionary significance they provide.

Response 4: Phylogenetic analysis has been modified for clarity and comprehension. 

5. The manuscript need to express in the introduction/ background section comments on current body of literature, after that, what are the gaps in our understanding of viral molecular evolution. The title of the manuscript is totally a mismatch from the contents.

Response 5: As per the suggestion of the reviewer, introduction section of the manuscript has been updated. 

6. Language editing is required to improve the quality of manuscript.

Response 6: In the light of reviewer’s comment, the manuscript has been revised thoroughly to improve the quality of language and eliminate grammatical and spelling mistakes.

---

## [Decision Letter · Decision Letter 1]

3 Dec 2021

Investigating Host-Virus Interaction Mechanism and Phylogenetic Analysis of Viral Proteins Involved in the Pathogenesis

PONE-D-21-33085R1

Dear Dr. Hassan,

We’re pleased to inform you that your manuscript has been judged scientifically suitable for publication and will be formally accepted for publication once it meets all outstanding technical requirements.

Kind regards,

Timir Tripathi, Ph.D.

Academic Editor

PLOS ONE

Additional Editor Comments (optional):

Reviewers' comments:

Reviewer's Responses to Questions

**Comments to the Author**

1. If the authors have adequately addressed your comments raised in a previous round of review and you feel that this manuscript is now acceptable for publication, you may indicate that here to bypass the “Comments to the Author” section, enter your conflict of interest statement in the “Confidential to Editor” section, and submit your "Accept" recommendation.

Reviewer #1: All comments have been addressed

Reviewer #2: All comments have been addressed

2. Is the manuscript technically sound, and do the data support the conclusions?

Reviewer #1: Yes

Reviewer #2: Yes

3. Has the statistical analysis been performed appropriately and rigorously? 

Reviewer #1: Yes

Reviewer #2: Yes

4. Have the authors made all data underlying the findings in their manuscript fully available?

Reviewer #1: Yes

Reviewer #2: Yes

5. Is the manuscript presented in an intelligible fashion and written in standard English?

Reviewer #1: Yes

Reviewer #2: (No Response)

6. Review Comments to the Author

Reviewer #1: Authors have revised the manuscript according to reviewers suggestions and comments. The manuscript in the present form may be accepted for publication.

Reviewer #2: (No Response)

7. PLOS authors have the option to publish the peer review history of their article (what does this mean?). If published, this will include your full peer review and any attached files.

Reviewer #1: No

Reviewer #2: **Yes: **RAJESH SINHA

---

## [Editor Report · Acceptance letter]

7 Dec 2021

PONE-D-21-33085R1 

Investigating Host-Virus Interaction Mechanism and Phylogenetic Analysis of Viral Proteins Involved in the Pathogenesis 

Dear Dr. Hassan:

I'm pleased to inform you that your manuscript has been deemed suitable for publication in PLOS ONE. Congratulations! Your manuscript is now with our production department. 

Kind regards, 

on behalf of

Dr. Timir Tripathi 

Academic Editor

PLOS ONE